# 6 nm super-resolution optical transmission and scattering spectroscopic imaging of carbon nanotubes using a nanometer-scale white light source

Xuezhi Ma [1], Qiushi Liu [1], Ning Yu [2], Da Xu [1], Sanggon Kim[2], Zebin Liu[3], Kaili Jiang [3], Bryan M. Wong [2,4], Ruoxue Yan[2,4 ✉] & Ming Liu[1 ✉]

Optical transmission and scattering spectroscopic microscopy at the visible and adjacent wavelengths denote one of the most informative and inclusive characterization methods in material research. Unfortunately, restricted by the diffraction limit of light, it cannot resolve the nanoscale variation in light absorption and scattering, diagnostics of the local inhomogeneity in material structure and properties. Moreover, a large quantity of nanomaterials has anisotropic optical properties that are appealing yet hard to characterize through conventional optical methods. There is an increasing demand to extend the optical hyperspectral imaging into the nanometer length scale. In this work, we report a super-resolution hyperspectral imaging technique that uses a nanoscale white light source generated by superfocusing the light from a tungsten-halogen lamp to simultaneously obtain optical transmission and scattering spectroscopic images. A 6-nm spatial resolution in the visible to near-infrared wavelength regime (415–980 nm) is demonstrated on an individual single-walled carbon nanotube (SW-CNT). Both the longitudinal and transverse optical electronic transitions are measured, and the SW-CNT chiral indices can be identified. The band structure modulation in a SW-CNT through strain engineering is mapped.

---

[1] Department of Electrical and Computer Engineering, University of California — Riverside, Riverside, CA 92521, USA. [2] Department of Chemical and Environmental Engineering, University of California — Riverside, Riverside, CA 92521, USA. [3] State Key Laboratory of Low Dimensional Quantum Physics and Department of Physics, Tsinghua University, 100084 Beijing, China. [4] Materials Science and Engineering program, University of California — Riverside, Riverside, CA 92521, USA. ✉email: rxyan@engr.ucr.edu; mingliu@ucr.edu

The colors of nanomaterials are determined by the optical absorption and scattering processes strongly correlated with their local optical and electronic structures, which can be radically different from the bulk. Single-walled carbon nanotubes (SW-CNTs), for example, comprise a family of more than 200 different structures that are characterized by different chiral indices, endeavored with distinct electronic structures[1,2], and known to show different colors as individuals[3–5]. On the contrary, in bulk, they are the darkest material that absorbs nearly all incident light[6,7]. Like other nanomaterials, SW-CNTs have electronic and optical properties closely related to the environmental influence, such as local strain, defects, dielectric screening, the quantum effect from particle size, etc. There is a strong drive for optical hyperspectral imaging techniques that can provide multidimensional information with nanometer resolution to decipher the local optical and electronic properties noninvasively.

Conventional optical spectroscopic microscopy has its spatial resolution restricted to hundreds of nanometers due to light diffraction. Although near-field scanning optical microscopy (NSOM) offers nanometer-scale resolution by using the plasmonic effect on an optical antenna to scan at the vicinity of the sample surface and has been widely applied absorption/scattering imaging via single-wavelength excitations[8–11], its applications in spectroscopy analysis in the visible region have been primarily restricted to inelastic light-matter interaction processes[12–14], such as tip-enhanced photoluminescence (TEPL) or Raman scattering (TERS), where sufficiently high signal-to-noise ratios can be achieved by removing the excitation light with a spectral filter. Recently, the NSOM-based nano-spectroscopic imaging has been demonstrated in the infrared (IR) regime, using spatially coherent light sources such as tunable mid-IR lasers[15–17], or a synchrotron radiation beam if a broad bandwidth is desired[18–20]. Extending the nano-spectroscopy imaging technique to the absorption and elastic scattering processes in the visible (VIS) and near-infrared (NIR) range will allow direct probing of band structures of a much wider variety of semiconductors with nanoscopic details, without requiring sample luminesces or advanced light sources.

Here, we report a strategy to extend the VIS-NIR scattering and transmission spectroscopic microscopy down to the nanometer length scale. The light from a tungsten-halogen lamp is compressed to the tip apex of a silver nanowire (AgNW) probe through high-external-efficiency broadband nanofocusing[21] to create a broad-spectrum ('white') point light source for nanoscale near-field sample illumination. The two-step nanofocusing process, as described previously[21], involves the mode coupling from the optical fiber (OF) to the AgNW waveguide and the adiabatic nanofocusing of the surface plasmon polaritons (SPPs) in the AgNW at its gradually narrowing tip. Since neither of the two steps requires spatial or spectral coherency in light, a tungsten-halogen light source can provide sufficient light intensity for hyperspectral imaging. We image both the transmission and scattering spectra of SW-CNTs with a 6-nm spatial resolution and analyze the longitudinal and transverse optical electronic transitions with this approach. The intrinsic electronic structure variation along a structured SW-CNT prepared by the local strain engineering is also studied.

## Results and discussion

**Design of the VIS-NIR hyperspectral NSOM.** The working mechanism of the nanoscale VIS-NIR hyperspectral microscopy can be considered as a dark-field NSOM configuration, as illustrated in Fig. 1a. The radially polarized SPP in the AgNW waveguide probe is quasi-adiabatically focused by the gradually narrowing tip (Fig. 1a zoom-in), forming a plasmonic hotspot at the tip apex (tip radius ~5 nm) with enhanced electric field

components in both parallel and perpendicular directions in respect to the sample surface. The far-field radiation pattern of the superfocused mode forms a radially polarized ring[22], with its 1st-order lobe as small as around 15° in the E-plane (Fig. 1a–i, inside of the dashed circle). The further k-space measurement confirms its radial polarization over the working wavelength range (Supplementary Note 3). As the probe approaches the sample surface, the superfocused electrical dipole at the tip apex starts to interact with its image dipole in the substrate, leading to the emerging of the 2nd-order lobes (Fig. 1a-ii, outside of the dashed circle). A k-space filter (NA = 0.7) is inserted into the optical path to block the low-k component (Fig. 1a-iii), leaving the high-k part to form a radially polarized ring pattern in the spectrometer image plane for spectrum analysis (Fig. 1a-iv).

The intensity profile along the azimuthal direction of the ring pattern is highly sensitive to the nanoscale optical anisotropy distribution. Conventionally, optical anisotropy results in the polarization variation in the transmitted or scattered light[5,23]. In a radially polarized beam, the polarization variation causes intensity variation along the azimuthal direction of the beam after focus[24–26]. Specifically, a SW-CNT placed along the x-direction, as shown Fig. 1b, has a strong depolarization effect due to the longitudinal electronic transition that weakens the x-direction far-field radiation (noted as $k_\parallel$). This variation can be measured by selecting the corresponding region of interest in the spectrometer camera (Fig. 1a-iv, red dashed area, noted as ROI$_\parallel$). Meanwhile, the longitudinal electronic transition generates longitudinal diploes that enhance the far-field radiation along the y-direction (defined as $k_\perp$), leading to an increase of light intensity in ROI$_\perp$. The spectroscopic information acquired from the two ROIs can reconstruct the nanoscale transmission and scattering images of the sample. As shown in Fig. 1c, the true-color pictures of two individual SW-CNTs are calculated from their spectra using the CIE 1931 color matching function. The separation distance between the two SW-CNTs is ~100 nm, well below Abbe's diffraction limit. It is worth noting that the low-k light from the probe can be scattered by sample surface roughness and become the major source of the noise, which influences the scattering image more severely than the transmission image due to the already weak signal level. Increasing the NA of the k-space filter can improve the image quality (Supplementary Note 4).

**Performance of the hyperspectral NSOM.** Figure 2a shows a set of spectrally resolved transmission images of a pristine (18, 16) SW-CNT on a thin quartz substrate, with a spectrum range from 415 nm (2.99 eV) to 980 nm (1.27 eV). Clear SW-CNT images emerge within three spectral regions around 510 nm, 730 nm, and 860 nm. Figure 2b shows the averaged transmission (red curve) and scattering spectra (blue curve) of the SW-CNT, acquired from ROI$_\parallel$ and ROI$_\perp$, respectively. The valley/peak positions indicate that they are from the longitudinal electronic transitions in a (18, 16) SW-CNT. Compared with the valleys in the transmission spectrum (508, 733, and 857 nm), the scattering spectrum peaks (490, 714, and 838 nm) are red shift by ~20 nm. This shift may originate from their mechanistic differences: the intensity loss in the absorption process is proportional to the imaginary part of the SW-CNT permittivity Im[$\varepsilon(\omega)$], whereas in the scattering process, the scattering strength scales quadratically with the induced electric momentum $|\varepsilon(\omega) - \varepsilon_{av}|^2$, where $\varepsilon_{av}$ is the averaged permittivity of the environment. The local maximum of these two equations may correlate with slightly different wavelengths.

Figure 2c shows the transmission and scattering hyperspectral images of the (18, 16) SW-CNT reconstructed from the spectra of ROI$_\parallel$ and ROI$_\perp$, respectively. The spatial resolution of both images can be estimated from a set of adjacent spectra across the SW-CNT. As shown in Fig. 2d, for the 2.3 nm-in-diameter SW-CNT, the

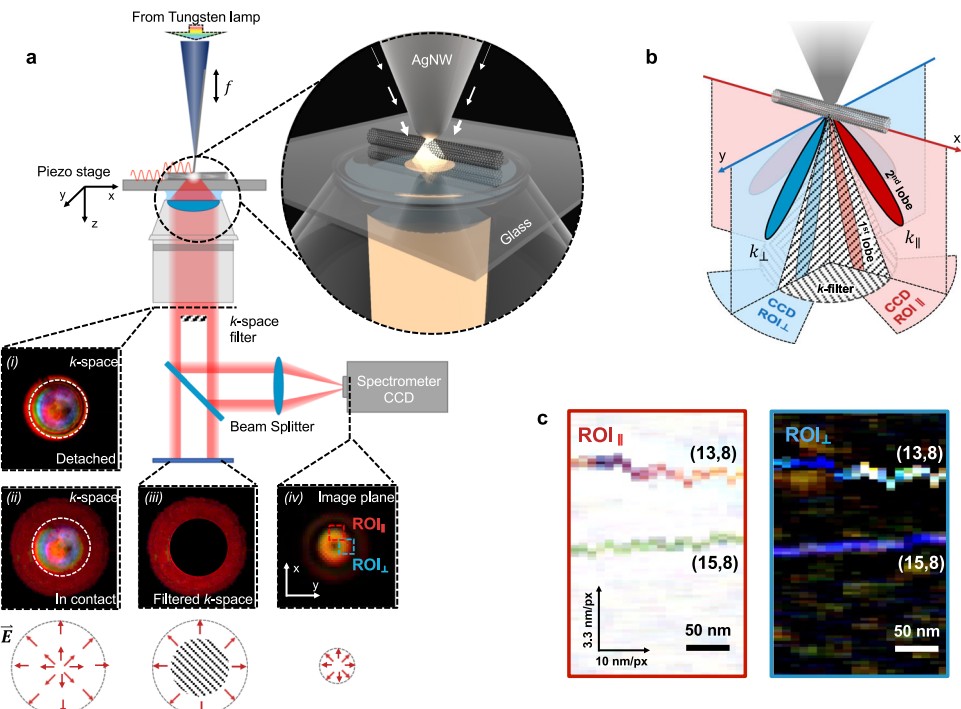

**Fig. 1 A scattering- and absorption-based nano-hyperspectral imaging design. a** Sketch of the experiment set-up and the k-space far-field radiation patterns when the probe detached from (i) and in contact with (ii) the substrate. Inset (iii) shows the pattern in (ii) with the low-k components (central portion) blocked by a k-space filter, and (iv) shows the filtered high k components (iii) focused to the image plane of a spectrometer for spectroscopy analysis. Bottom insets: polarizations of the above images. **b** The radially polarized far-field radiations have azimuthal-dependent components (noted as $k_{\parallel}$ and $k_{\perp}$) that are linked with different regions in the image plane (noted as ROI$_{\parallel}$ and ROI$_{\perp}$). **c** The transmission (left panel) and scattering (right panel) true-color images of two SW-CNTs, separated by ~100 nm from each other, acquired from ROI$_{\parallel}$ and ROI$_{\perp}$, respectively. The chiral indices are (13,8) and (15,8), respectively. At each pixel (with a pixel step of 3.3 nm for vertical direction and 10 nm for lateral direction), one acquisition with 0.5 s integration time was recorded to obtain the two spectrums from different ROIs simultaneously.

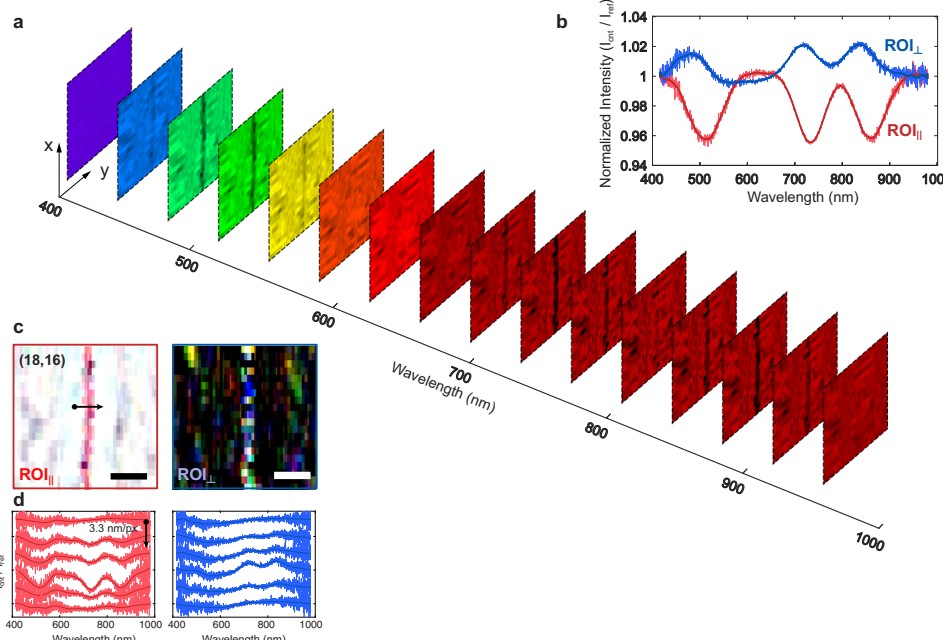

**Fig. 2 Transmission spectroscopic images of an (18, 16) SW-CNT sample. a** Slices from the hyperspectral data show the transmission images ROI$_{\parallel}$ of an (18,16) SW-CNT at different excitation wavelengths with a 35 nm interval. **b** Representative spectra of the SW-CNT acquired from ROI$_{\parallel}$ and ROI$_{\perp}$, respectively, by accumulating the spectra from 20 points along the SW-CNT. The reference spectrum is taken from the same sample with the same probe. **c** True-color images of the SW-CNT, constructed by converting the hyperspectral images into the RGB (red, green, and blue) channels using the CIE 1931 color matching functions. Scale bar: 50 nm. Pixel step: 3.33 nm (horizontal) and 10 nm (vertical). **d** Six adjacent near-field spectra across the SW-CNT along the black arrow in **c**, collected from ROI$_{\parallel}$ (left panel) and ROI$_{\perp}$ (right panel), respectively. The scanning step is 3.3 nm per pixel. The integration time is 0.3 s per spectrum. The total light intensity delivered by the probe is roughly 30 nW.

spatial resolution is ~6 nm for the 510 nm transmission valleys, calculated from fitting its 2D Gaussian surface in the wavelength-displacement space (Supplementary Note 5). This resolution is roughly the same as the AgNW tip radius (~5 nm).

**Resolving the anisotropic optical properties.** SW-CNT is one of the ideal quasi-one-dimensional systems and has highly aniso-tropic optical properties. Due to their one-dimensional char-acteristic, SW-CNTs have strong optical transitions when the incident light polarization is parallel to their axes, which has been intensively investigated through inelastic (e.g. photoluminescence excitation spectroscopy[27–29]) and elastic scattering measurements (e.g. Rayleigh scattering microscopy[1,30,31]), including near-field imaging measurements[32]. The Rayleigh scattering with a per-pendicular polarization, however, is difficult to characterize due to the small scattering cross section and has only been investi-gated theoretically[33]. The superfocused nanoscale light source contains strong parallel and perpendicular electric fields and can simultaneously measure longitudinal and transverse optical transitions. We illustrate the band structures and indices of a semiconducting SW-CNT near the Fermi level in Fig. 3a, b to explain the characteristic features of different transitions. The excitation from a valence band $n_v$ to a conduction band $n_c$ has the band index difference $\triangle n = n_c - n_v$, which, as required by the selection rule, needs to satisfy $\triangle n = 0$ or $\triangle n = \pm 1$ for excitation polarization parallel or perpendicular to the SW-CNT axis. These two scenarios have different influences on the far-field radiation patterns, distinguishable from the two ROI spectra. Specifically, the longitudinal transition (Fig. 3a) attenuates the horizontal electrical component ($\vec{E}_\parallel$) of the superfocused plas-monic hotspot, leading to a reduced far-field intensity in ROI$_\parallel$. On the other hand, the longitudinal excitons have dipole-like radiations, with radiation energy concentrating at the SW-CNT's radial direction. Consequently, the light intensity along with $k_y$ increases. The spectra acquired from ROI$_\parallel$ and ROI$_\perp$ have opposite characteristics (valleys vs. peaks) at the longitudinal transition frequencies (shaded green), noted as the $E_{ii}$ transitions (e.g., $E_{22}^s$, $E_{33}^s$, $E_{44}^s$... the superscript $s$ denotes that they are semiconductor-type SW-CNTs) in Fig. 3d–f. The transverse optical transitions $E_{ij}$ ($E_{12}^s$, $E_{21}^s$, $E_{13}^s$, $E_{31}^s$..., shaded orange), how-ever, attenuate the superfocused plasmonic hotspot through interacting with the vertical electric field ($\vec{E}_\perp$), which reduces the overall radiation intensity and leaves valleys in both spectra. The relative peak intensity variation for different transverse transi-tions can be estimated through the band structures calculated by the first-principles density function theory[34,35] (DFT, see "Methods" section). We take an (8, 6) SW-CNT as an example. The DFT calculated band structure (Fig. 3c) indicates that both the $E_{12}^s$ (or $E_{21}^s$) and $E_{22}^s$ transitions have quasi-direct bandgaps with a small mismatch in momentum (~3% and 7% of the Bril-louin zone, respectively), which can be efficiently excited by the superfocused light at the probe apex and appear in both trans-mission and scattering spectra as evident valleys/peaks. The $E_{13}^s$ transition is across an indirect bandgap with a more considerable momentum mismatch (~40% of the Brillouin zone), making it more challenging to excited. Nonetheless, since its unit length (2.6 nm) is comparable with the probe tip radius (~5 nm), this momentum mismatch can still be partially compensated by the sharp tip. Therefore, $E_{13}^s$ transition gives a shallow yet measurable valley in both spectra of Fig. 3d.

Similar characteristics for longitudinal and transverse transi-tions can be identified in all SW-CNT scan results. Figure 3g, h are the Kataura plots for $\triangle n = 0$ and $\triangle n = \pm 1$ transitions[36], which help visualize the correlation between the optical transition

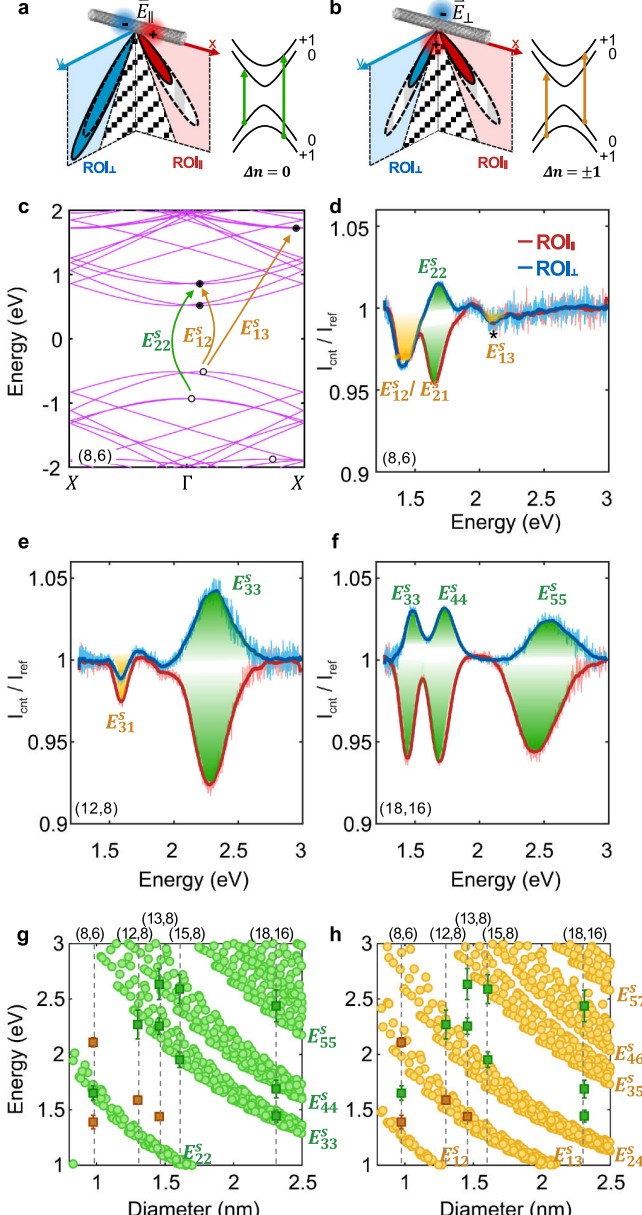

**Fig. 3 Optical transmission and scattering spectra from the longitudinal and transverse optical transitions of semiconductor SW-CNTs. a** Left panel: the longitudinal optical absorption reduces the far-field intensity in ROI$_\parallel$ through absorption and increases the power in ROI$_\perp$ through scattering. Right panel: energy bands of a semiconductor SW-CNT near the Fermi level with band indices. The allowed transitions for longitudinal optical absorption (green arrows) require $\triangle n = 0$. **b** Left panel: when the transverse absorption occurs, the far-field radiation intensities are reduced in both ROIs. Right panel: allowed transitions for the transverse optical absorption (orange arrows) require $\triangle n = \pm 1$. **c** Electronic DFT band structure and **d** the corresponding ROI$_\parallel$ and ROI$_\perp$ spectra of an (8, 6) SW-CNT. **e, f** The representative spectra from two other SW-CNTs, with chiral indices of (12, 8) and (18, 16), respectively. **g, h** Kataura plots showing the energies of longitudinal transitions ($\triangle n = 0$) and transverse transitions ($\triangle n = \pm 1$) in different semiconductor SW-CNTs. The green (orange) squares indicate the calculated longitudinal (transverse) transitions. Error bars indicate the peaks/valleys widths.

energies and chiral indices of the SW-CNTs. Kataura plot indicates the relationship between the bandgap energies in a SW-CNT and its diameter[36], which can be experimentally determined by the measured radial-breathing mode (RBM) frequencies

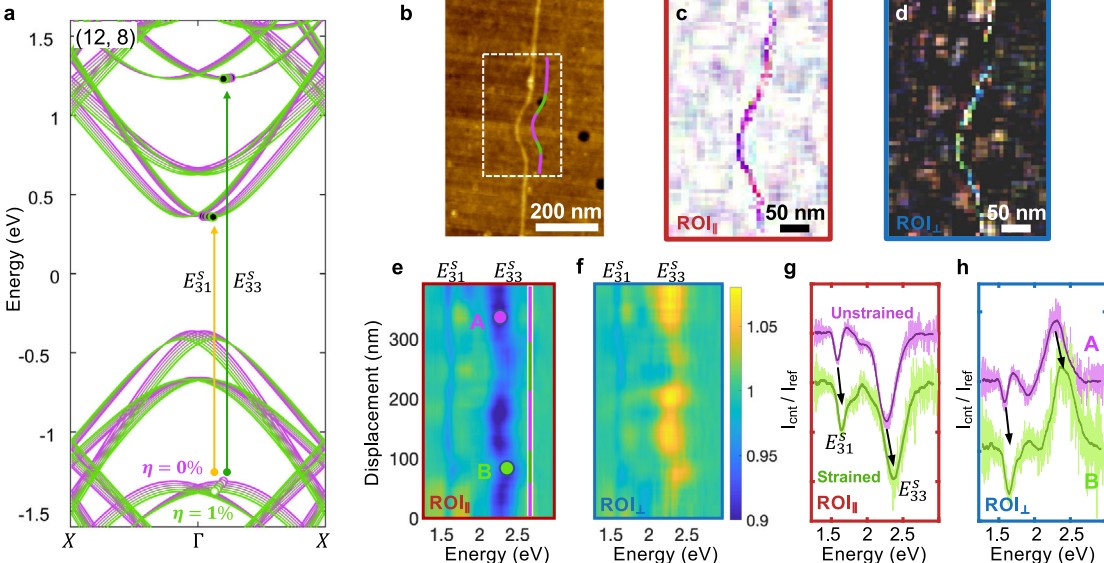

**Fig. 4 Optical transitions in a strained SW-CNT. a** Electrical DFT band structure of a (12, 8) SW-CNT under different uniaxial strain from 0% (purple line) to 1% (green line). **b** The AFM topographical image of a curved (12, 8) SW-CNT, created by the AFM-based nanomanipulation on a straight CNT. The purple-green line is the eye-guide for strained (green) and unstrained (purple) regions along the nanotube. **c, d** The true-color images of the dashed region in **b**, acquired from ROI$_\parallel$ and ROI$_\perp$, respectively. **e, f** depict the spectra taken from all points on the SW-CNT, plotted against their displacement. **g, h** show the spectra averaged from 10 adjacent data points picked from the unstrained region A and strained region B.

(Supplementary Note 7). The corresponding optical transition energies are calculated from the spectra and marked by the green ($\triangle n = 0$) or orange ($\triangle n = \pm1$) squares, referring to the longitudinal and transverse excitations. We notice that the transverse-excitation transitions are more prominent in small-diameter SW-CNTs. In contrast, in the large-diameter ones, the features are either buried by the strong signals from longitudinal excitations or vanish due to large momentum mismatches.

**Resolving the strain-induced spametial inhomogeneity**. The super-resolution transmission and scattering spectroscopic imaging can also diagnose the local inhomogeneity in band structures. Strain engineering is an effective way to tune the band structures and tailor the electronic and optical performance of semiconductors[37,38]. With the nano-spectroscopic imaging technique described here, we can examine the strain-induced spatial modulation of band-structure along a single SW-CNT. Figure 4b is the topographical image of a strained (12, 8) SW-CNT, prepared through an AFM nanomanipulation method[39,40] with a super-aligned SW-CNT sample (Supplementary Note 6). DFT calculation (see "Methods" section) in Fig. 4a reveals that in a (12, 8) SW-CNT, 1% uniaxial strain ($\eta$) can modulate the $E_{31}^s$ and $E_{33}^s$ transitions by around 3%, with both transitions showing strain-induced blue shifts. The transmission and scattering hyperspectral images (Fig. 4c, d) of the strained segment of the SW-CNT (white dashed box) show slight color variation, indicating the spectral shifts. Figure 4e, f depict the measured transmission and scattering spectra, plotted against their displacements along the SW-CNT. Blue shifts in $E_{33}^s$ and $E_{31}^s$ transitions are observed at both edges of the crescent kink (marked with green lines), demonstrating that the strain is concentrated on the transition regions instead of the middle of the crescent. This is possibly because the friction between the SW-CNT and the substrate was too weak to pin the entire kinked region, and the middle segment of the kinked region relaxed over time. At the high-strain transition regions, both the longitudinal transition $E_{33}^s$ and the transverse transition $E_{31}^s$ blue-shift by ~3% (Fig. 4g, h), equivalent to ~1% of strain according to the DFT calculation. We also notice that the $E_{31}^s$ peak intensity slightly increased under strain (Fig. 4f, g), which may result from the

reduction in momentum mismatch between the corresponding valance and conduction bands, as shown in the DFT calculation (Fig. 4a).

In summary, the super-resolution optical transmission and scattering spectroscopic imaging method provides a powerful tool for optical property characterization at the nanometer scale, in our case offering direct insights into the strain-induced bandgap modulation along a SW-CNT. By improving the power density of the light source, such as switching the tungsten-halogen lamp to a supercontinuum white-light laser, high-speed imaging up to a few frames per second is possible[41]. The fiber-based nature also offers the flexibility to perform the measurement under a cryogenic environment. Our technique pushes the spatial resolution of VIS-NIR imaging from sub-micrometer into the nanometer regime and can potentially shine the light on catalysis, quantum optics, nanoelectronics, and more.

## Methods

**Density functional theory calculations of SW-CNT**. The electronic properties of the large (8,6) and (12,8) SW-CNTs were calculated using an all-electron 6–21G(d) polarized basis set in conjunction with the B3LYP hybrid functional. We have specifically chosen to use this theory level since prior work has shown that the hybrid B3LYP functional accurately predicts experimental bandgaps and electronic structures of SW-CNTs compared to other semi-local functionals. Full geometry optimizations (i.e., for both the atomic positions and the unit cell) for the unstrained (8,6) and (12,8) SW-CNTs were carried using periodic boundary conditions without imposing any symmetry constraints. The number of atoms in the (8,6) and (12,8) SW-CNTs is 296 atoms/4100 orbitals and 304 atoms/4256 orbitals, respectively. With the unstrained geometry of the (12,8) SW-CNT converged, a series of constrained B3LYP/6-21 G(d) calculations were carried out at different uniaxial strains to produce Fig. 4a in the main text. Electronic band structures for both the (8,6) and (12,8) SW-CNTs were obtained and resolved with an extremely fine mesh of 100 k-points along the Brillouin zone.

**NSOM measurement**. A tapping-mode tuning-fork configuration[42,43] (frequency f in Fig. 1a is 32.7 kHz) was adopted for the experiment on a commercial NSOM module (Nanonics, model Multiview 2000), which is integrated with an inverted optical microscope (Nikon, model Eclipse Ti-U) and an optical spectrometer (Princeton Instrument, model Acton 2300) for spectrum analysis. The light provided by a tungsten-halogen lamp (Thorlabs, model SLS201L) is coupled into the other end of the fiber probe through a fiber optic collimator. The far-field radiation from the AgNW tip is collected by an oil-immersion objective (NA = 1.3), and its radial polarization is

confirmed by polarization analysis. A k-space filter (NA = 0.7) is inserted into the optical path to block the low-k component, leaving the high-k part sent to the side camera port for spectrum collection. The probe tip image in the spectrometer camera is first checked under the zero-order reflection mode to confirm its doughnut shape. The two ROIs can then be selected. More details about the spectrometer setting and CNT sample preparation can be found in Supplementary Notes 1 and 2.

## Data availability

The data that support the plots within this paper and other findings of this study are available from the corresponding authors upon reasonable request.

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

## Acknowledgements

M.L. acknowledges support by the National Science Foundation (no. 1654746 and 1810453). R.Y. acknowledges support by NSF award no. 1654794. DFT calculations by B.M.W. were supported by the U.S. Department of Energy, Office of Science, Basic Energy Sciences, TCMP Program, under Award No. DE-SC0022209. K.J. acknowledges support by the National Natural Science Foundation of China (no. 51727805 and 51788104).

## Author contributions

M.L. and R.Y. initiated the project, designed the experiments, and supervised the research. X.M. fabricated and characterized the probe, performed the transmission and scattering spectroscope measurements. X.M. and Q.L. analyzed the data. X.M performed the k-space measurements with assistance from Q.L. and D.X.; N.Y. and S.K. synthesized the sharp-tip AgNW. Z.L. and K.J. provided the single-walled CNT sample. X.M. transferred the CNTs with assistance from Z.L.; B.W. performed the DFT calculations. All authors contributed to discussion and manuscript preparation.

## Competing interests

The authors declare no competing interests.
