## [Peer Review File · Nature Communications]

REVIEWER COMMENTS

Reviewer #1 (Remarks to the Author):

This paper reports a novel UV-VIS absorption and scattering imaging technique, with record-high spatial and spectral resolution. By combining their breakthrough superfocusing technique with the dark-field NSOM configuration, the authors succeeded in demonstrating a full-color nano-imaging of nanomaterials with sub-5 nm spatial resolution, more than two orders better than the conventional UV-VIS technique. Astonishingly, no laser or any coherent light source was required to achieve such performance: a tungsten lamp is enough. The authors also take advantage of the radial symmetry of the fundamental SPP mode in a silver nanowire waveguide and observe the anisotropic scattering from single-walled carbon nanotubes. For the first time, the scattering from perpendicularly polarized excitons is reported. The authors also mapped the strain-induced bandgap variation along a CNT, which is important for not only scientific researches but also the semiconductor industry. Considering all these highly appealing achievements, I highly recommend its publication in Nature Communication.

I have only one minor concern about the title. Its current form is less attractive compared with a recent paper published in Science Advances a few days ago, titled "White nanolight source for optical nanoimaging", even though the current work is much more solid and sound than the other. The authors should come up with a more generalized title to attract a broader range of readers.

Reviewer #2 (Remarks to the Author):

This paper concerns the use of a nanofocusing technique to achieve nanometer-scale spatially-resolved absorption imaging of single-walled carbon nanotubes. The analysis of absorption and scattering in CNTs, taking advantage of the interesting polarization properties of nanofocusing, is carefully-considered. However, the authors claim that nanoscale optical approaches generally are based on vibrational spectroscopies, and their approach, of visible/near-IR absorption and scattering approach is therefore novel and valuable to understanding bandgap heterogeneity. Unfortunately, they appear to have overlooked a great deal of research in the field on different approaches to nanoscale electronic absorption, and therefore do not place their work in the appropriate context.

For example, they cite only Kim et al., Nature Photonics volume 13, pages636–643(2019) (reference 17 in the text) in the context of nanofocusing. While this is the closest experimental description of their nanofocusing implementation, other studies have been performed to investigate the polarization properties that the authors discuss in detail in the manuscript, most importantly Esmann et al., "k-space imaging of the eigenmodes of sharp gold tapers for scanning near-field optical microscopy", Beilstein J Nanotechnol. 2013; 4: 603–610. Furthermore, other geometries and launching designs have also been used for similar nanofocusing-based visible hyperspectral nano-imaging, e.g. "Highly efficient plasmonic tip design for plasmon nanofocusing in near-field optical microscopy", Umakoshi et al., Nanoscale 2016, 8, 5634-5640.

Furthermore, there have been considerable advances in employing other approaches than nanofocusing to perform nanoscale band gap heterogeneity imaging. While visible scattering-based near-field optical microscopy has been developed, it is not extensively used, but photoinduced force microscopy can overcome many of its limitations. See, for example, Yoon et al., "Nanoscale Imaging and Spectroscopy of Band Gap and Defects in Polycrystalline Photovoltaic Devices", Nanoscale. 2017 Jun 14; 9(23): 7771–7780. This type of approach has even been employed to study transient absorption of carbon nanotubes, "Nanoscale Excitation Dynamics of Carbon Nanotubes Probed with Photoinduced Force Microscopy", Kim et al., Phys. Chem. C 2020, 124, 21,

11694–11700. While the authors of the current manuscript could argue that their approach of using a cheap and readily-available tungsten-halogen lamp is less specialized than expensive laser systems, overall the novelty factor is not what they claim. None of these previous important developments were cited in the manuscript.

Finally, while the study of the shift in local absorption with strain in CNTs has not to my knowledge been investigated before, it is also close to other similar studies that are not appropriately cited: "Tip-enhanced nano-Raman analytical imaging of locally induced strain distribution in carbon nanotubes", Yano et al., Nature communications 4 (1), 1-7. For these reasons I suggest that the paper lacks sufficient novelty for publication in Nature Communications.

Reviewer #3 (Remarks to the Author):

Ma et al. demonstrate a creative approach to performing nanoscale absorption and scattering spectroscopy. The authors demonstrate the ability to perform nanoscale polarization anisotropy, darkfield scattering, and absorption spectroscopy on a single one-dimensional carbon nanotube with very sub-diffraction spatial resolution. The principal advance in their approach is the ability to measure the transverse optical absorption of a single carbon nanotube, which, to my knowledge is the first of its kind (as the authors point out). Except for minor issues, the work is technically sound, and the conclusions are well-supported by the data and theory presented in the manuscript. In a few cases that are minor (and summarized below), the conclusions could be strengthened by a clearer presentation and additional data analysis, and in a few instances, the authors should provide a bit more detail on their methodology. I believe these points are addressable and should not prohibit publication.

My major concern with the manuscript is its significance and novelty. A very similar technique was published last year by Verma et al. [Science Advances 3, eaba4179 (2019)]. In that work, (which is not cited in the current manuscript), the authors perform similar nano-optical absorption spectroscopy on carbon nanotubes with comparable resolutions (the current manuscript presents a better spatial resolution by $\sim 3\times$). The central difference between the current work and the prior work by Verma et al. is that Ma et al. demonstrate the additional ability to measure the transverse absorption of a single carbon nanotube. However, this new capability seems to be restricted to 1D structures (and perhaps just carbon nanotubes) and a clear case is not presented on how it would apply to other material systems. Furthermore, although this work is the first demonstration of measuring the transverse absorption of a single carbon nanotube (to the best of my knowledge), a clear case for how such characterization provides new insight into carbon nanotubes is also not made, especially in the context of other ensemble polarization anisotropy studies of aligned carbon nanotubes (e.g. Advanced Materials 18, 1166 (2006), PRL 98, 147402 (2007), and citations thereof). For these reasons, despite the technical soundness of the current work, I question whether it constitutes a significant advance in the areas of nano-optical materials characterization or carbon nanotubes and thus whether the work would be of broad interest to either community. The manuscript appears to be better suited for a more specialized journal in its current form.

A few minor, but less serious issues should be addressed:

1. What is the significance of the "true" color rendering of the hyperspectral data? Or why display the data in such a non-quantitative form when the aim is to visualize the bandgap heterogeneity? In the current form, it is difficult/impossible to determine how much the bandgap changes in those plots.
2. The ROIs for the absorption and scattering measurements need to be selected and aligned to the orientation of the carbon nanotubes. The authors should describe how they perform this important alignment step.
3. Claiming a sub-5 nm spatial resolution is dubious when the pixel size is only 3.3 nm. Strictly speaking, from the Nyquist criteria, the smallest resolution demonstratable in the image is $2\times$ the

pixel size in the image which would be 6.6 nm.

4. It would be nice to see a comparison of the imaging with and without the k-space filter employed to extract only the nonlinear response in the system. How does the nonlinear aspect of the response modify the interpretation of the absorption and scattering spectra (i.e., the relative intensities of the valleys and peaks)? This latter question is important from a materials characterization standpoint.

5. The spectra reported in Figs. 2, 3, and 4 are reported as "Intensity" on the y-axis. This labeling appears to be misleading – the spectra are relative changes in intensity, $\Delta I/I$. It should also be noted where/how the normalization spectrum was acquired. And presumably, this normalization spectrum is tip-dependent.

6. In Fig. 4, the locations used for the averaging of the strained and unstrained regions are not specified – are they continuous regions of the nanotube?

7. In Fig. 3, the Kataura plots should be described in a bit more detail for a broad audience. Methods describing how they were created are not included in the methods sections.

8. Many grammatical errors throughout the text should be corrected to improve readability.

Response to Reviewer #1:

Comment: This paper reports a novel UV-VIS absorption and scattering imaging technique, with record-high spatial and spectral resolution. By combining their breakthrough superfocusing technique with the dark-field NSOM configuration, the authors succeeded in demonstrating a full-color nano-imaging of nanomaterials with sub-5 nm spatial resolution, more than two orders better than the conventional UV-VIS technique. Astonishingly, no laser or any coherent light source was required to achieve such performance: a tungsten lamp is enough. The authors also take advantage of the radial symmetry of the fundamental SPP mode in a silver nanowire waveguide and observe the anisotropic scattering from single-walled carbon nanotubes. For the first time, the scattering from perpendicularly polarized excitons is reported. The authors also mapped the strain-induced bandgap variation along a CNT, which is important for not only scientific researches but also the semiconductor industry. Considering all these highly appealing achievements, I highly recommend its publication in Nature Communication. I have only one minor concern about the title. Its current form is less attractive compared with a recent paper published in Science Advances a few days ago, titled “White nanolight source for optical nanoimaging”, even though the current work is much more solid and sound than the other. The authors should come up with a more generalized title to attract a broader range of readers.

Response: We are extremely grateful for Reviewer 1’s very positive feedback and comments, and the endorsement of our new NSOM technique.

Reviewer 1 reminds us of a recently published paper on a similar topic in *Science Advances*, to which we would like to clarify the online publication date (*June 3rd*) was after our submission date (*May 15th*). Our current title is indeed relatively narrow and may fail to attract broad interests, and per Reviewer 1’s suggestion, we would like to change the title to “6 nm super-resolution optical transmission and scattering spectroscopic imaging of carbon nanotubes using a nanometer-scale white light source”.

Response to Reviewer #2:

Comment: This paper concerns the use of a nanofocusing technique to achieve nanometer-scale spatially-resolved absorption imaging of single-walled carbon nanotubes. The analysis of absorption and scattering in CNTs, taking advantage of the interesting polarization properties of nanofocusing, is carefully-considered. However, the authors claim that nanoscale optical approaches generally are based on vibrational spectroscopies, and their approach, of visible/near-IR absorption and scattering approach is therefore novel and valuable to understanding bandgap heterogeneity. Unfortunately, they appear to have overlooked a great deal of research in the field on different approaches to nanoscale electronic absorption, and therefore do not place their work in the appropriate context.

For example, they cite only Kim et al., Nature Photonics volume 13, pages636–643(2019) (reference 17 in the text) in the context of nanofocusing. While this is the closest experimental description of their nanofocusing implementation, other studies have been performed to investigate the polarization properties that the authors discuss in detail in the manuscript, most importantly Esmann et al., "k-space imaging of the eigenmodes of sharp gold tapers for scanning near-field optical microscopy", Beilstein J Nanotechnol. 2013; 4: 603–610. Furthermore, other geometries and launching designs have also been used for similar nanofocusing-based visible hyperspectral nano-imaging, e.g. "Highly efficient plasmonic tip design for plasmon nanofocusing in near-field optical microscopy", Umakoshi et al., Nanoscale 2016, 8, 5634-5640.

Response: We thank Reviewer #2 for the careful review and the suggestions. The Reviewer mentions that “*the authors claim that nanoscale optical approaches generally are based on vibrational spectroscopies, and their approach, of visible/near-IR absorption and scattering approach is therefore novel and valuable...*”. It was not our intention to diminish other works and we apologize if we have given the Reviewer such impression. We would like to clarify though that we did not actually say or mean that “*nanoscale optical approaches are based on vibrational spectroscopies*”, which would indeed be false. What we wrote in the original manuscript was “*its (NSOM) applications in **spectroscopy** analysis at the **visible** region mainly targets **inelastic** light-matter interaction processes*”. What we meant by this statement is: when working in the visible regime, NSOM has been restricted to single wavelength measurements (including scattering/absorption imaging) with no spectroscopy analysis capacity, with the exception of **inelastic** processes, including **BOTH** photoluminescence (PL) and vibrational spectra, both of which can generate PL or Raman spectra from a single wavelength light source. We did acknowledge in the next sentence that for the mid-IR regime, elastic spectroscopy analysis (absorption/scattering) with NSOM is prevalent. However, it holds true that the same had not been accomplished in the visible regime due to the difficulty in generating a nanoconfined white light source. We do not feel that this original statement was an over claim, however, given the

complexity of the issue discussed, we have rephrased the statement as follows to avoid any potential confusion:

(Updated manuscript, changed part is in blue) *“Although near-field scanning optical microscopy (NSOM) offers nanometer-scale resolution by using the plasmonic effect on an optical antenna to scan at the vicinity of the sample surface and has been widely applied absorption/scattering imaging via single-wavelength excitations, its applications in spectroscopy analysis in the visible region have been primarily restricted to inelastic light-matter interaction processes, such as tip-enhanced photoluminescence (TEPL) or Raman scattering (TERS), where sufficiently high signal-to-noise ratios can be achieved by removing the excitation light with a spectral filter.*

Next, we would like to address the two references suggested by Reviewer 2.

The first suggestion, "k-space imaging of the eigenmodes of sharp gold tapers for scanning near-field optical microscopy", Beilstein J Nanotechnol. 2013; 4: 603–610, was actually cited in our previous work, *Nature Photonics*, 13, 636, which **focused on introducing a nanofocusing technique**, which has heavy discussion on the role of polarization in nanofocusing, and therefore aligns well with the Beilstein J Nanotechnol reference. Under the citation number restriction, we did not feel the need to reference it in the current manuscript because this manuscript did not repeat mode or polarization-related discussions, which is what the Beilstein J Nanotechnol reference focused on, and cited other more relevant works from Dr. Lienau's group.

The second suggestion, *"Highly efficient plasmonic tip design for plasmon nanofocusing in near-field optical microscopy"*, Umakoshi et al., *Nanoscale* 2016, 8, 5634-5640, was not cited because it was a single-wavelength measurement and not hyperspectral imaging, which we felt not closely relevant to our present work.

We did not mean to overlook or omit these two references and both are now included in the updated manuscript. And we would like to thank Reviewer 2 for holding the manuscript to a high standard.

*Comment: Furthermore, there have been considerable advances in employing other approaches than nanofocusing to perform nanoscale band gap heterogeneity imaging. While visible scattering-based near-field optical microscopy has been developed, it is not extensively used, but photoinduced force microscopy can overcome many of its limitations. See, for example, Yoon et al., "Nanoscale Imaging and Spectroscopy of Band Gap and Defects in Polycrystalline Photovoltaic Devices", *Nanoscale*. 2017 Jun 14; 9(23): 7771–7780. This type of approach has even been employed to study transient absorption of carbon nanotubes, "Nanoscale Excitation Dynamics of Carbon Nanotubes Probed with Photoinduced Force Microscopy", Kim et al., *Phys. Chem. C* 2020, 124, 21, 11694–11700. While the authors of the current manuscript could argue*

that their approach of using a cheap and readily-available tungsten-halogen lamp is less specialized than expensive laser systems, overall the novelty factor is not what they claim.

None of these previous important developments were cited in the manuscript.

Response: We appreciate the discussion Reviewer 2 brought up on the relation between direct absorption/scattering imaging and pulse-induced force microscopy (PiFM). PiFM is an emerging technique that combines the plasmonic enhancement for the optical response with the high mechanical sensitivity from a conventional AFM. As we understand it, in a PiFM measurement, a single wavelength pump pulse laser is focused on the probe tip to locally heat the sample, inducing thermal expansion in the latter that can be picked up by the AFM as the signal. The output is the force signal encoded in the AFM probe frequency/phase shift, which do not provide direct spectral information. The absorption images shown in the two references mentioned by Reviewer 2, were single wavelength mapping of the sample. With a tunable laser, the Nanoscale (2017) reference were able to do force mapping under 3 and NSOM mapping with 5 discrete excitation wavelengths, and the JPCC (2020) reference (Figure 5) showed PiFM amplitude mapping of intertwining bundles of SWCNTs with a single wavelength laser at three different pump-probe delay time. With any single wavelength excitation method, regardless of what the output signal is (AFM amplitude as in PiFM or optical response as in conventional NSOM), one can only reconstruct a hyperspectral image (i.e., each pixel contains the information of an entire spectrum) by sweeping the wavelength of a continuously tunable laser at each pixel, which would be tremendously time-consuming and technically challenging.

It is also worth mentioning that SWCNT bundles imaged in the JPCC (2020) reference give a stronger absorption (from the already-intensive pulsed laser) than the single SWCNT in our experiment. There hasn't been prior reports of an individual SWCNT measured by scattering/absorption NSOM due to the very weak signal of a single SWCNT (e.g. Kamaras et al. in *"Scattering near-field optical microscopy on metallic and semiconducting carbon nanotube bundles in the infrared"* (Phys. Status Solidi B, 255, 2413)).

In contrast, the technology reported in this manuscript measure the entire visible scattering/absorption spectrum (400 - 1000 nm) from **each pixel** in <1 second and are sensitive enough for bandgap inhomogeneity imaging in a single SWCNT. The **spectral resolution** (<1 nm) is only limited by the spectrometer and its grating, and the **spatial resolution** is ~6 nm, much higher than the PiFM method. We strongly believe that the major novelty factor of this method is enabling hyperstroscopy scattering/absorption nanoimaging, which revealing much finer details in the spectral domain and spatial domain of nanoscale material inhomogeneity than any currently available optical methods.

That being said, we would like to sincerely thank Reviewer 2 for noticing the other major advantage of our work: “*approach of using a cheap and readily-available tungsten-halogen lamp is less specialized than expensive laser systems*”. The reported method can replace costly coherent (tunable) laser sources with a cheap tungsten-halogen lamp (Stabilized Tungsten IR Light Source, 450 - 5500 nm, Thorlabs SLS202L) that provides **broadband spectra** (415nm-980nm) excitation for both absorption and scattering measurements. Despite of the **absence** of spatial or spectral coherence (necessary for homodyne or heterodyne detection in a mono-color NSOM) in such broadband lightsource, a spatial resolution ~ 6 nm has already been achieved with the nanofocusing method.

Finally, it is worth noting that the work published in *Phys. Chem. C* (**May. 5. 2020**) was very close to our submission date (**May. 15. 2020**), and thus we could not include it during submission. It is cited in the revised manuscript.

Comment: Finally, while the study of the shift in local absorption with strain in CNTs has not to my knowledge been investigated before, it is also close to other similar studies that are not appropriately cited: "Tip-enhanced nano-Raman analytical imaging of locally induced strain distribution in carbon nanotubes", Yano et al., Nature communications 4 (1), 1-7. For these reasons I suggest that the paper lacks sufficient novelty for publication in Nature Communications.

Response: We thank Reviewer 2 for acknowledging the novelty of the absorption and scattering spectrum study on strained CNTs. We fully agree that one can use the vibrational peaks in Raman spectroscopy to tell if a CNT is strained. However, the information that can be acquired is rather limited. For example, Raman cannot precisely distinguish the chiral indices of a SWCNT or be used to evaluate the strain-induced spatial variations of its band structure, which is the key information provided by our work.

There are hundreds of commonly used single-walled CNTs with different chiral indices and types (semiconducting or metallic). Although there are ways to roughly determinate the CNT diameter using the radial-breathing mode (RBM) of a CNT:

$$\tilde{\nu}_{RBM} = \frac{A}{d_t} + B$$

where $A=223.5\text{cm}^{-1}\text{nm}$ and $B=12.5\text{cm}^{-1}$ (Ref^[1]), determining the chiral index of a SWCNT, or even its CNT type is far more difficult. For example, an (18,16) and a (17,17) CNT share similar diameters, 2.308nm and 2.307nm, respectively. The corresponding RBM peaks are 109.34cm^{-1} and 109.38cm^{-1} , which are almost indistinguishable under the influence of surrounding environments. On the other hand, their electronic properties are entirely different, as one is metallic,

and the other is semiconducting. This is a common issue for CNTs with similar chiral indices. Therefore, even if the RBM information can be provided by **TERS** (which has not been reported to the best of our knowledge, presumably due to the large fluorescent noise from the metallic probe at low wavenumbers), it is still fundamentally challenging to determine the chirality and study band structure with Raman. In contrast, the method reported in this manuscript can provide a comprehensive chirality, bandstructure and strain analysis of any individual SWCNT from the broadband absorption and scattering spectra measurement. The high spatial resolution in the strain CNT mapping image obtained in this also provides detailed information on how the band-gap of CNT responds to tensile stress and vary spatially (**Figure 4**). These experimental data is verifiable by and agree well with DFT simulations.

In addition, the **simultaneous** absorption and scattering spectra measurement is essential for distinguishing the parallel and perpendicular transitions in a SWCNT, which is another **UNIQUE and NOVEL** capability brought to the table by our technique. Parallel transitions have opposite signs for the same feature in the two spectra, while the perpendicular transition shows the same signs. Parallel transitions correspond to the same indices for the initial and final states ($\Delta n = 0$, e.g., E_{11} , E_{22} , E_{33} , etc.) while the perpendicular transitions with $\Delta n = \pm 1$ provide the information of E_{12} , E_{21} , E_{13} , etc. To our best knowledge, our work is the **first DIRECT measurement** of the perpendicular transition part of the Kataura plot. Other than our work, we could only find one **indirect** measurement (*“Selection-rule breakdown in plasmon-induced electronic excitation of an isolated single-walled carbon nanotube”*^[2] Nat. Photonics 2013), where some data points that cannot be categorized by the $\Delta n = 0$ transition are concluded as the $\Delta n = \pm 1$ transition. In contrast, the prevailing method for CNT absorption measurements^[3], i.e., the spatial modulation spectroscopy (SMS) technique, only measures the parallel transition ($\Delta n = 0$), as summarized in Table 1 (adapted from ref ^[3]). The much smaller scattering/absorption cross-section of the perpendicular absorption makes it more difficult to detect and has not been previously reported. Our DFT simulations (**Figure 3c** and **Figure 4a**) also confirm that some of the perpendicular absorptions correspond to indirect band gaps with a large momentum mismatch. The lack of a lightning-rod enhancement and an indirect bandgap could also offer insight on why the perpendicular absorption has not been experimentally observed, even in CNT bundles, until this work.

Table 1. Absorption measurement on SWCNT. Please note that only parallel transitions ($\Delta n = 0$) are reported (shadowed area). Table adapted from ref 4.

SWNT	Type	Energy transition	S_{ii} (eV)	$C_{abs}^{S_{ii}}$ (nm ² nm ⁻¹)	Peak absorption ($\times 10^{-17}$ cm ² per carbon atom)		ν (meV)
					$C_{abs}^{S_{ii}}$	C_{abs}	
d=2 nm Free-standing	Type I (22,6)	S ₃₃	1.66	0.60	2.5	3.1	55
		S ₄₄	2.21	0.20	0.8	1.4	51
		S ₅₅	2.61	0.24	1.0	1.6	102
d=1.83 nm Free-standing	Type I (14,13)	S ₃₃	1.93	0.64	2.9	3.5	50
		S ₄₄	2.27	0.30	1.4	1.9	52
		S ₅₅	3.06	0.31	1.4	2.0	102
d=1.83 nm On substrate	Type I (14,13)	S ₃₃	1.87	0.22	1.0	1.7	273
		S ₄₄	2.21	0.48	2.2	2.9	244
		S ₅₅	3.04	0.39	1.8	2.5	272
d=2.24 nm Free-standing	Type II	S ₃₃	1.64	0.57	2.1	2.8	61
		S ₄₄	1.79	0.63	2.4	3.0	71
		S ₅₅	2.50	0.08	0.3	1.0	194
		S ₆₆	2.62	0.26	1.0	1.6	80
d=2.49 nm Free-standing	Type II	S ₃₃	1.65	0.91	3.0	4.1	66
		S ₄₄	1.79	1.13	3.7	4.9	80
		S ₅₅	2.50	0.09	0.3	1.5	135
		S ₆₆	2.63	0.52	1.7	2.8	116

Response to Reviewer #3:

Comment: Ma et al. demonstrate a creative approach to performing nanoscale absorption and scattering spectroscopy. The authors demonstrate the ability to perform nanoscale polarization anisotropy, darkfield scattering, and absorption spectroscopy on a single one-dimensional carbon nanotube with very sub-diffraction spatial resolution. The principal advance in their approach is the ability to measure the transverse optical absorption of a single carbon nanotube, which, to my knowledge is the first of its kind (as the authors point out). Except for minor issues, the work is technically sound, and the conclusions are well-supported by the data and theory presented in the manuscript. In a few cases that are minor (and summarized below), the conclusions could be strengthened by a clearer presentation and additional data analysis, and in a few instances, the authors should provide a bit more detail on their methodology. I believe these points are addressable and should not prohibit publication.

Response: We sincerely thank Reviewer 3 for the careful review and strong recommendation for publication. We also appreciate the detailed suggestions, which tremendously help to improve the quality of the present work. The Reviewer suggests adding more details on the methodology, which are now included in the revised manuscript and SOI. Since most of the probe fabrication-related details have been discussed in our previous papers (*Nature Photonics*, 13(9), pp.636-643; *Nanoscale* 11 (16), 7790-7797), here we mainly focus on the microscope setup and data acquisition section, which also happen to be the most exciting yet confusing part of the work.

Comment: My major concern with the manuscript is its significance and novelty. A very similar technique was published last year by Verma et al. [Science Advances 3, eaba4179 (2019)]. In that work, (which is not cited in the current manuscript), the authors perform similar nano-optical absorption spectroscopy on carbon nanotubes with comparable resolutions (the current manuscript presents a better spatial resolution by ~3x). The central difference between the current work and the prior work by Verma et al. is that Ma et al. demonstrate the additional ability to measure the transverse absorption of a single carbon nanotube. However, this new capability seems to be restricted to 1D structures (and perhaps just carbon nanotubes) and a clear case is not presented on how it would apply to other material systems.

Response: We thank Reviewer 3 for bringing up this issue, which gives us the opportunity to explain. We also recently noticed this *Science Advances* paper from Dr. Umakoshi and Dr. Verma, two leading researchers of the field. We would like to humbly point out that its online publication date was actually **June 9th 2020** (not 2019), a few weeks **after** our initial submission to *Nature Communications* (on **May 15th 2020**). Therefore, we could not cite it during our submission. This paper has now been included in our revised manuscript.

There are some significant differences between our current work and the *Sci. Adv.* paper. **Firstly**, we map the spatial variation of an individual SWCNT band structure, which is very

challenging due to the extremely low signal level of a single SWCNT. Actually, in our NSF proposal (NSF 1654746, started in 2016 [4]), where we first proposed this idea, we already listed “*full-color NSOM*” to elucidate “*the band structure distributions in individual single-walled carbon nanotubes*” as a major challenge of the project. This measurement later became the most time-consuming part of our project, but its results also indisputably prove the capability of the method. **Secondly**, although measuring the transverse absorption sounds like an trivial increment to the parallel absorption, it is actually extremely challenging technically as well. Not only is the scattering cross-section for transverse absorption much smaller than the parallel absorption (due to its indirect bandgap and the lack of lightning-rod enhancement), but distinguishing these two absorptions requires a high-quality superfocused TM_{01} mode over all visible wavelengths to serve as the light source. Even after interacting with the sample, the majority of the far-field radiation still needs to maintain its radial polarization, which can then form a ring in the spectrometer CCD, and allow us to allocate the region of interests (ROI) to correlate different ROIs with the anisotropic properties of the sample. All these efforts together lead to the high spatial resolution (i.e., ~ 6 nm, details in our response to minor question #3), which is 5 times better than the *Sci. Adv.* work (31nm), with a cheap tungsten lamp as the light source.

We fully understand Reviewer 3’s concern about the universality of the new technique. SWCNTs are standard samples to benchmark a new instrument due to their naturally reduced dimensions (1D), which is why we used SWCNT, and so did the *Sci. Adv.* paper and many more. Besides, a strained SWCNT has a large kink that might be regarded as a quasi-2D sample. The nice result verifies the compatibility of this technique.

Comment: Furthermore, although this work is the first demonstration of measuring the transverse absorption of a single carbon nanotube (to the best of my knowledge), a clear case for how such characterization provides new insight into carbon nanotubes is also not made, especially in the context of other ensemble polarization anisotropy studies of aligned carbon nanotubes (e.g. Advanced Materials 18, 1166 (2006), PRL 98, 147402 (2007), and citations thereof). For these reasons, despite the technical soundness of the current work, I question whether it constitutes a significant advance in the areas of nano-optical materials characterization or carbon nanotubes and thus whether the work would be of broad interest to either community. The manuscript appears to be better suited for a more specialized journal in its current form.

Response: We would like to thank Reviewer 3 for endorsing that we carry out the first observation of the “transverse absorption” in a SWCNT. A concern raised here is about the impact of the measurement. We would like to point out that the transverse absorption is more than merely a complementary measurement to the parallel absorption and has long been pursued by the CNT community, although all reported results only studied the parallel transitions up until now [5-8]. The optical transition measurement on SWCNTs can be used to study strong many-body interactions, the efficiency of photon-electron generations, and quantized optical transitions, thus enabling a

broad range of applications from photonics, optoelectronics, and photovoltaics. In some scenarios, such as Förster resonance energy transfer in SWCNT-quantum dot systems for photovoltaics where short-range interactions dominate, the contribution of transverse absorption could be comparable to the parallel absorption and it is essential that we have tools to study these absorptions. Our experimental design fills in this gap and provides more opportunities for related research.

The transverse absorption also provides a important path to understand the indirect bandgap transitions in SWCNTs. The absorption spectrum of a CNT shows the parallel transitions with $\Delta n = 0$, which provide information on van hove singularities with the same subscript indices (e.g. E_{11} , E_{22} , E_{33} , etc). The perpendicular transitions with $\Delta n = \pm 1$ provide information on the van hove singularities with different indices, such as E_{12} , E_{21} , E_{13} , etc. As shown by the DFT simulations (**Figure 3c** and **Figure 4a**), the parallel transitions are mostly direct band gaps (even though they may simply be close to the Γ point). The transverse absorption, however, has indirect bandgaps in most of these cases. The absorption strength can reflect the magnitude of the corresponding mismatch in the Brillouin zone. For example in a (8,6) SWCNT (**Figure 3c**), the E_{13} transition is an indirect bandgap with a larger momentum mismatch ($\sim 60\%$ of the Brillouin zone), while the E_{12} has only a 3% mismatch. Therefore, the E_{12} valley is much deeper than the E_{13} valley (**Figure 3d**).

Although the transverse absorption is essential, it has not been reported previously, presumably due to the following reasons: (1) the parallel absorption can benefit from the lightning-rod effect of a SWCNT and thus leads to larger absorption signals, and (2) most of the transverse absorptions are between indirect band gaps, which has a large momentum mismatch that cannot be compensated by the momentum carried by a free-space photon. This momentum mismatch can be compensated by an NSOM tip's geometry, leading to its emerging in our measurement. To our best knowledge, our work is the **first** one to show the Kataura plots for **both** parallel and perpendicular transitions, with an excellent resolution and small error bars.

A few minor, but less serious issues should be addressed:

We thank Reviewer 3 for these **minor concerns**, motivating us to conduct additional experiments and data analysis. We respond to these questions below.

Comment: 1. What is the significance of the “true” color rendering of the hyperspectral data? Or why display the data in such a non-quantitative form when the aim is to visualize the bandgap heterogeneity? In the current form, it is difficult/impossible to determine how much the bandgap changes in those plots.

Response: We fully agree with Reviewer 3 that compressing the spectrum data into RGB channels unavoidably loses information. With that being said, we actually take this colorful visualization as a chance to quickly explain to **general readers** about the capability and importance of near-field optical imaging. In our daily life, what we perceive as the color of an object is based on its optical far-field scattering/radiation, where the information can be really smeared up, however, extending to the quantum region, the colors of an object can be completely different and can only be captured with optical near-field imaging. The carbon nanotube is a good example. CNT powders are pitch black, but we show that it is counterintuitively colorful in the nearfield. We want to be straight to the point in the manuscript by showing colorful SWCNT images.

In addition, we fully agree that the detailed spectrum can deliver more information in a quantitative way, including the bandgap, absorption, scattering peaks, etc. Therefore, we provide detailed spectra for both aligned and strained carbon nanotubes in *Figure 3* and *Figure 4*. We also carried out large-scale DFT calculations to show how the band-structure changes under tensile strain (*Figure 4*).

Comment: 2. The ROIs for the absorption and scattering measurements need to be selected and aligned to the orientation of the carbon nanotubes. The authors should describe how they perform this important alignment step.

Figure R1. SEM images of aligned-CNT array transferred on glass substrate with gold marker.

Response: The Reviewer raises a keen question about the complexity of the proposed technique. **Ideally**, only basic optical alignment operations are required to align the light source image into the spectrometer slit. **Practically**, we use aligned SWCNT arrays to facilitate assigning the two regions of interest.

We use the aligned SWCNTs provided by collaborator Dr. Kaili Jiang, who has the synthesis method described in their prior work (*Nature Catalysis*, 1(5), pp.326-331). The aligned SWCNT arrays were grown on a quartz crystal substrate and then transferred onto a thinner quartz substrate by a standard PMMA-assisted wet transfer method. We transferred the CNTs array onto the glass with gold markers (See **Figure R1**), which allows us to put SWCNTs along a specific direction for scanning (e.g. x-direction in Figure 1). With the SWCNT orientation fixed, we can define two regions of interest in the spectrometer camera.

Figure R2 shows scanning probe images, acquired from the side camera port of the inverted optical microscope that is integrated with the NSOM scanning module (Nanonics Multiview 2000). **Figure R2a** is a high-resolution image acquired by a color CCD, showing a precise doughnut shape with a radial polarization (proven in the SOI). **Figure R2b** is the image of the same probe acquired by a spectrometer camera (Princeton Instrument Acton 2300), with the grating set at zero-order reflection mode. The white dashed lines in **Figure R2a** indicate the location of the front slit, which cut the doughnut ring into half to improve the spectral resolution; otherwise, the ring can broaden the peaks in the spectrum (by a few pixels). The real slit size in **Figure R2b** is around 100 μm , so the spectral resolution is not compromised.

Figure R2. Probe tip images acquired by (a) a high-resolution CCD camera and (b) spectrometer camera (Princeton Instruments Acton 2300), respectively. The entrance slit of the spectrometer is marked by white dashed lines. (c), Ideally, more ROIs can be picked for a better polarization analysis.

The doughnut pattern occupies about 7 horizontal lines (7 pixels along the vertical direction) in our spectrometer CCD camera. We group them into two regions of interest, labeled as ROI_{\parallel} and ROI_{\perp} . We could do this grouping because we already have the SWCNT fixed along the x-axis (corresponding to the vertical direction in the spectrometer reference frame). We set only two groups because the usb2.0 cable used by the spectrometer has a low data transfer rate, as a 2×1400 data array already consumed around 1.5 seconds, even though our exposure time was only 0.3 second.

With that being said, in the ideal case (**Figure R2c**), the ROIs can be divided finer if the spectrometer CCD has a smaller pixel size and higher data transfer rate. With more ROIs defined, the predetermination of sample orientations is no longer required.

A corresponding discussion is included in the updated SOI.

Comment: 3. Claiming a sub-5 nm spatial resolution is dubious when the pixel size is only 3.3 nm. Strictly speaking, from the Nyquist criteria, the smallest resolution demonstratable in the image is 2x the pixel size in the image which would be 6.6 nm.

Response: We would like to thank Reviewer 3 for raising this concern. We claimed sub-5 nm resolution because many of the spectra look like delta-function distribution compared with the neighboring spectra. Meanwhile, we were also not sure if this situation still obeys the Nyquist criteria. The Nyquist criterion is applied in the signal processing field with single spatial sampling. In our case, however, an additional dimension (in the frequency domain) participates. This extra dimension can be regarded as multiple measurements on the same sample, a similar scenario as the localization method can get resolution better than the pixel size.

Therefore, we suggest using the 2D Gaussian fitting to determine the spatial resolution. In the full-color CNT mapping, we continuously picked out a series of spectra that perpendicularly across the (18, 16) SWCNT, as shown in **Figure R3**. We performed a 2D Gaussian fitting (in both the displacement and wavelength direction) and found the spatial resolution at **6 nm** for the short-wavelength peak/valley at 510 nm.

A corresponding discussion is included in the updated SOI.

Figure R3. 2D-Gaussian fitting for the spatial resolution calculation. **a~c:** absorption spectra. **d~f:** scattering spectra. This (18, 16) SWCNT is the same one used in Figure 2. The spatial resolution is 6 nm for the short-wavelength valley/peak in **a** and **d**.

Comment: 4. It would be nice to see a comparison of the imaging with and without the k -space filter employed to extract only the nonlinear response in the system. How does the nonlinear aspect of the response modify the interpretation of the absorption and scattering spectra (i.e., the relative intensities of the valleys and peaks)? This latter question is important from a materials characterization standpoint.

Response: We thank Reviewer 3 for this suggestion. The k -space filter turns out to be a critical component in the measurement. Because the 1st-order lobe of the far-field radiation from the TM_0 mode in a silver nanowire has a small radiation angle, as shown in **Figure R4a**, the far-field radiation (without any contact between the probe and substrate) becomes a strong background in the measurement if it is not filtered. This far-field radiation originates from the TM_0 mode scattered by the AgNW tip. Since the overall momentum mismatch between the TM_0 mode and free-space light is small ($\sim 10\%$), the scattering angle is therefore small as well. This scattered light does not contain any sample-related information and cannot be used for imaging purposes.

We have tried measurements without the k -space filter, but no CNT features could be identified.

Reviewer 3 asked about the influence of the nonlinear aspect of the response to the absorption and scattering spectra. Since we use a tapping mode AFM (with the oscillation peak-to-peak amplitude around 10 nm), we actually work at a small tip-to-sample distance with a relatively linear response, and we cannot have a constant tip-to-sample distance to examine this nonlinear effect. However, inspired by the Reviewer's question, we did check the influence from another nonlinear aspect: k -space filter size. We re-scanned the (18, 16) CNT used in Figure 2 with a larger k -space filter (NA=0.9, the one used in the manuscript was 0.7), and surprisingly found that the scattered signals in the ROI_{\perp} disappeared. The absorption images (ROI_{\parallel}) remain with a similar quality, and **Figure R4** shows the new results. This phenomenon can be explained by the reasons below:

- 1) Unlike absorption, the CNT scatters the light to a narrow solid angle, and the scattering signal can only be detected within a small range of k -space. At $NA > 0.9$, the scattering signal hardly exists. This radiation pattern might differ from the doughnut-shaped dipole radiation pattern (the parallel excitons can be considered as an electric dipole, with its polarization along the CNT axis^[3]). We suspect the lack of high-NA information should come from the influence of the silver tip, which may serve as the reflector in a Yagi-Uda antenna.
- 2) The absorption images seem to show a better signal-to-noise ratio, which may come from the fact that high-NA signals have less influence from other noise, such as the Mie scattering from sample contaminations.

A corresponding discussion is included in the updated SOI.

Figure R4. Comparison of the scanning images on an (18,16) CNT with (a) a smaller k -space filter (NA = 0.7) and (b) a larger k -space filter (NA = 0.9).

Comment: 5. The spectra reported in Figs. 2, 3, and 4 are reported as “Intensity” on the y-axis. This labeling appears to be misleading – the spectra are relative changes in intensity, $\Delta I/I$. It should also be noted where/how the normalization spectrum was acquired. And presumably, this normalization spectrum is tip-dependent.

Response: We appreciate Reviewer 3 for point out this issue. The word “Intensity” is indeed misused when actually referring to I_{cnt}/I_{ref} . It has been changed in the revised manuscript.

The normalization spectra were acquired by normalizing the raw spectrum from each pixel with a reference spectrum acquired from a glass-only area, using the same probe on the same sample. Since the SWCNT distribution is sparse on the substrate, it is relatively easy to find a reference spectrum. This method can also avoid the tip-dependent variation in the superfocused light-source spectrum, which is unavoidable since it can be influenced by many aspects, such as the AgNW diameter, the cleanliness of the cleaved optical fiber butt, and even the light bulb working condition.

Comment: 6. In Fig. 4, the locations used for the averaging of the strained and unstrained regions are not specified – are they continuous regions of the nanotube?

Response: We thank the Reviewer for raising this question. The 10 data points are continuous points from the strained and unstrained regions, separately. We label these two regions in the revised manuscript.

Comment: 7. In Fig. 3, the Kataura plots should be described in a bit more detail for a broad audience. Methods describing how they were created are not included in the methods sections.

Response: We thank Reviewer 3's suggestion. The Kataura plots should be explained to a broad audience, especially those unfamiliar with carbon nanotubes. We include a short discussion on Kataura plots in the revised manuscript, as listed below:

“Kataura plot indicates the relationship between the band gap energies in a single walled carbon nanotube (SWCNT) and its diameter^[9]”.

Comment: 8. Many grammatical errors throughout the text should be corrected to improve readability.

Response: We apologize for the grammatical errors, even though we have carefully gone through the manuscript word by word before the submission. We have carefully rechecked the manuscript and revised many sentences, and hope it reads better this time.

References:

- 1 S. M. Bachilo, M. S. Strano, C. Kittrell, R. H. Hauge, R. E. Smalley & R. B. Weisman. *Structure-assigned optical spectra of single-walled carbon nanotubes*. **science** 298, 2361-2366, (2002).
- 2 M. Takase, H. Ajiki, Y. Mizumoto, K. Komeda, M. Nara, H. Nabika, . . . K. Murakoshi. *Selection-rule breakdown in plasmon-induced electronic excitation of an isolated single-walled carbon nanotube*. **Nature Photonics** 7, 550-554, (2013).
- 3 J.-C. Blancon, M. Paillet, H. N. Tran, X. T. Than, S. A. Guebrou, A. Ayari, . . . F. Vallée. *Direct measurement of the absolute absorption spectrum of individual semiconducting single-wall carbon nanotubes*. **Nature Communications** 4, 2542, (2013).
<https://doi.org/10.1038/ncomms3542>
- 4 M. Liu. *NSF Award*. (2016).
https://www.nsf.gov/awardsearch/showAward?AWD_ID=1654746&HistoricalAwards=false
- 5 K. Liu, J. Deslippe, F. Xiao, R. B. Capaz, X. Hong, S. Aloni, . . . F. Wang. *An atlas of carbon nanotube optical transitions*. **Nature Nanotechnology** 7, 325-329, (2012).
<https://doi.org/10.1038/nnano.2012.52>
- 6 M. S. Hofmann, J. T. Glückert, J. Noé, C. Bourjau, R. Dehmel & A. Högele. *Bright, long-lived and coherent excitons in carbon nanotube quantum dots*. **Nature Nanotechnology** 8, 502-505, (2013).
<https://doi.org/10.1038/nnano.2013.119>
- 7 K. Liu, X. Hong, Q. Zhou, C. Jin, J. Li, W. Zhou, . . . F. Wang. *High-throughput optical imaging and spectroscopy of individual carbon nanotubes in devices*. **Nature Nanotechnology** 8, 917-922, (2013).
<https://doi.org/10.1038/nnano.2013.227>

- 8 F. Yao, C. Liu, C. Chen, S. Zhang, Q. Zhao, F. Xiao, . . . K. Liu. *Measurement of complex optical susceptibility for individual carbon nanotubes by elliptically polarized light excitation. **Nature Communications** 9, 3387, (2018).*

<https://doi.org/10.1038/s41467-018-05932-9>

- 9 H. Kataura, Y. Kumazawa, Y. Maniwa, I. Umezu, S. Suzuki, Y. Ohtsuka & Y. Achiba. *Optical properties of single-wall carbon nanotubes. **Synthetic metals** 103, 2555-2558, (1999).*

REVIEWERS' COMMENTS

Reviewer #2 (Remarks to the Author):

The authors have responded carefully and in detail to the previous comments in my review and those of the other reviewers. I believe their changes have strengthened and clarified the manuscript and I am satisfied with their responses, and support publication of the paper.

I do wish to clarify for the authors that photoinduced force microscopy (PiFM), as developed by Wickramasinghe and performed by the company Molecular Vista, and photothermal microscopy (sometimes called PTIR or AFM-IR, etc.) from Anasys/Bruker are separate techniques. To my knowledge, both can be performed in broadband implementation but this may have only been demonstrated in the infrared.

Reviewer #3 (Remarks to the Author):

In my initial review of the work, I had two significant concerns. The first concern was with the novelty of the work, especially in light of the recent work by Verma et al. As the authors correctly and politely pointed out, I regrettably misidentified the year that it was published. The authors have addressed this recent work adequately in the new manuscript. Most certainly, their technique complements these recent results, and in a few cases in terms of light sources, polarization sensitivity, and resolution achieved, the work achieves notable advances.

My second significant concern was regarding the potential impact of the manuscript. Although the technique achieves the aforementioned notable advancements, it is specialized to SW-CNTs. The authors still do not make a strong case that it can be adapted to materials that are less-ordered and more difficult to align with respect to the collection optics. On the other hand, the results present the first measurements (that I know of) of optical absorption of SW-CNTs with polarization oriented orthogonal to the SW-CNT, which is noteworthy. I am disappointed to see that the importance of these states is not discussed in more detail. The authors make a decent case in their response that did not make it into the manuscript (although I question the argument that weak transitions are important in processes such as FRET). So, the technique still appears to be a specialized way to measure weak transitions in SW-CNTs.

However, the bottom line is that overall, I find the work to be technically sound, the technique to be a potentially interesting and transformative way to study other material systems, and the study of SW-CNTs to be a reasonable proof-of-concept. The technique has potential and I look forward to seeing it applied to other materials. For this latter reason, I think the manuscript passes the bar for Nature Communications and recommend it for publication.

All of my minor concerns were adequately addressed.

Response to Reviewer #2:

The authors have responded carefully and in detail to the previous comments in my review and those of the other reviewers. I believe their changes have strengthened and clarified the manuscript and I am satisfied with their responses, and support publication of the paper.

We thank Reviewer #2 for the recommendation for publishing our manuscript.

I do wish to clarify for the authors that photoinduced force microscopy (PiFM), as developed by Wickramasinghe and performed by the company Molecular Vista, and photothermal microscopy (sometimes called PTIR or AFM-IR, etc.) from Anasys/Bruker are separate techniques. To my knowledge, both can be performed in broadband implementation but this may have only been demonstrated in the infrared.

We thank Reviewer #2 for the clarification.

Response to Reviewer #3:

In my initial review of the work, I had two significant concerns. The first concern was with the novelty of the work, especially in light of the recent work by Verma et al. As the authors correctly and politely pointed out, I regrettably misidentified the year that it was published. The authors have addressed this recent work adequately in the new manuscript. Most certainly, their technique complements these recent results, and in a few cases in terms of light sources, polarization sensitivity, and resolution achieved, the work achieves notable advances.

We thank Reviewer #3 for the endorsement of the novelty of this work.

My second significant concern was regarding the potential impact of the manuscript. Although the technique achieves the aforementioned notable advancements, it is specialized to SW-CNTs. The authors still do not make a strong case that it can be adapted to materials that are less-ordered and more difficult to align with respect to the collection optics. On the other hand, the results present the first measurements (that I know of) of optical absorption of SW-CNTs with polarization oriented orthogonal to the SW-CNT, which is noteworthy. I am disappointed to see that the importance of these states is not discussed in more detail. The authors make a decent case in their response that did not make it into the manuscript (although I question the argument that weak transitions are important in processes such as FRET). So, the technique still appears to be a specialized way to measure weak transitions in SW-CNTs.

However, the bottom line is that overall, I find the work to be technically sound, the technique to be a potentially interesting and transformative way to study other material systems, and the study of SW-CNTs to be a reasonable proof-of-concept. The technique has potential and I look forward to seeing it applied to other materials. For this latter reason, I think the manuscript passes the bar for Nature Communications and recommend it for publication.

All of my minor concerns were adequately addressed.

We thank Reviewer #3 for the recommendation for publishing our manuscript. We tried to add more measurements on other low-dimension materials (e.g. MoS₂) but the stringent working requirements during the pandemic slows down the progress. More related works may come in the future.